# Artificial Neural Network Modelling of the Effect of Vanadium Addition on the Tensile Properties and Microstructure of High-Strength Tempcore Rebars

**DOI:** 10.3390/ma15113781

**Published:** 2022-05-25

**Authors:** Woonam Choi, Sungbin Won, Gil-Su Kim, Namhyun Kang

**Affiliations:** 1R&D Center, Dongkuk Steel, 70 Geonposaneop-ro, 3214beon-gil, Nam-gu, Gyeongsangbuk-do, Pohang 37874, Korea; woonam.choi@dongkuk.com (W.C.); sungbin.won@dongkuk.com (S.W.); gilsu.kim@dongkuk.com (G.-S.K.); 2Department of Materials Science and Engineering, Pusan National University, Busan 46241, Korea

**Keywords:** Tempcore, high strength rebar, V-alloyed rebar, CCT diagram, V(C, N) precipitation, artificial neural network, yield strength

## Abstract

In high-strength rebar, the various microstructures obtained by the Tempcore process and the addition of V have a complex effect on the strength improvement of rebar. This study investigated the mechanism of strengthening of high-strength Tempcore rebars upon the addition of vanadium through artificial neural network (ANN) modelling. Various V contents (0.005, 0.072 and 0.14 wt.%) were investigated, and a large amount of bainite and V(C, N) were precipitated in the core of the Tempcore rebar in the high-V specimens. In addition, as the V content increased, the number of these fine precipitates (10–30 nm) increased. The precipitation strengthening proposed by the Ashby–Orowan model is a major contributing factor to the yield-strength increase (35 MPa) of the Tempcore rebar containing 0.140 wt.% V. The ANN model was developed to predict the yield and tensile strengths of Tempcore rebar after the addition of various amounts of V and self-tempering at various temperatures, and it showed high reproducibility compared to the experimental values (R-square was 93% and the average relative error was 2.6%). ANN modelling revealed that the yield strength of the Tempcore rebar increased more significantly with increasing V content (0.01–0.2 wt.%.) at relatively high self-tempering temperatures (≥530 °C). These results provide guidelines for selecting the optimal V content and process conditions for manufacturing high-strength Tempcore rebars.

## 1. Introduction

With the development of the construction industry, high-strength steels are being increasingly used in high-rise and long-span structures [1,2,3,4]. Rebars, which account for most primary construction materials, require high strength to ensure safety. High-strength rebars offer several advantages, such as reduced reinforcement ratio, reduced cost of reinforcement placement, improved workability and safety of buildings and increased service life owing to enhanced corrosion resistance [5,6,7]. Furthermore, the use of high-strength rebar conserves resources and is environment-friendly as it reduces greenhouse gas generation, which is inevitable in steel production by reducing the use of rebars.

Several studies have been conducted to develop high-strength steel for building structures. Among them, optimising the content of micro-alloy elements, such as V, Nb and Ti, is known to be effective in achieving high-strength steels [8,9,10,11,12,13]. Fine carbonitrides produced by the addition of micro-alloy elements improve the strength of steel by interfering with dislocation movement owing to the pinning effect and solute drag effect according to the solid solubility of the micro-alloy elements. In particular, V precipitates delay austenite recrystallisation at low deformation temperatures [8] and are effective in precipitation strengthening owing to their low solubility compared with other micro-alloy carbonitrides [9]. It is essential to understand the recrystallisation behaviour of V-containing steel according to the process conditions and micro-alloys because the grain refinement and precipitation strengthening are substantially affected by the austenite recrystallisation behaviour.

Recrystallization behaviour depends on several conditions in the steel manufacturing process, such as the recrystallisation temperature, deformation temperature, deformation amount, deformation rate and cooling rate, thereby affecting the final strength [14]. However, previous studies on strength improvement by V carbonitride have mainly focused on flat-rolled products, and studies on rebars are limited [15,16]. Rebars have a substantially high hot deformation rate, therefore producing a recrystallisation behaviour different from that of flat-rolled products. Moreover, a controlled cooling process called Tempcore is performed to achieve high strength. Various microstructures and micro-alloys obtained by the Tempcore process have a complex effect on the strength of rebars [17,18]. Therefore, to understand the strengthening mechanism of high-strength rebars, a systematic study on the complex relationship between micro-alloys, micro-structural factors, cooling conditions and mechanical properties is required.

Recently, artificial neural network (ANN) modelling, which is based on learning the relationships between the input and output parameter for complex problems, has been applied to predict and analyse various material phenomena [19,20,21,22,23,24,25,26]. The most remarkable feature of ANN modelling is the understanding of relationships using input and output data, and it can be implemented if there is sufficient learnable parameter data. Hong et al. [19] predicted the tensile properties of ferrite-pearlite steel using alloying elements and micro-structural factors, and determined the relative importance of each factor. Hosseini et al. [20] developed a model to predict the mechanical properties of transformation-induced plasticity steel subjected to heat treatment. Khalaj et al. [21] developed an ANN model to predict the Vickers hardness and yield strength of low-carbon steels based on previously reported data. Finally, Çetinel et al. [22] predicted the phase fractions of Tempcore rebars with a diameter of 18 mm subject to different quenching durations. Churyumov et al. [23] developed an ANN model to determine the steel flow stress with high accuracy in a wide range of elemental concentrations of high-alloy and corrosion-resistant steels. Honysz [24] used ANN to predict the chemical concentration of common alloying elements based on the mechanical property values of ferritic stainless steels. These studies successfully defined the unclear input–output relationship of the parameters of steels using ANNs. However, no ANN-based studies have been conducted on the effects of micro-alloys and controlled cooling parameters on the strength of high-strength Tempcore rebars.

This study investigated the effect of V addition on the strength of high-strength Tempcore rebars with a yield strength of 700 MPa. A continuous cooling transformation (CCT) diagram was used to analyse the micro-structural changes of the Tempcore rebar, and the effect on the mechanical properties was investigated through the analysis of the precipitate behaviour. The mechanical properties of rebars with various V contents and Tempcore process conditions were predicted using ANN modelling, and the influence of each factor was identified. This study provides a fundamental understanding of the mechanical properties of high-strength Tempcore rebars and guidelines for selecting V contents and Tempcore process conditions for manufacturing high-strength rebars in the steel industry.

## 2. Materials and Methods

### 2.1. Materials and Experimental Procedures

Table 1 lists the chemical compositions and mechanical properties of the rebar used in this study. To confirm the effect of V addition, rebars with various V contents (0.005, 0.072 and 0.14 wt.%) were used. Specimen RB V2 had the largest amount of V content, RB V0 had the smallest content and RB V1 had a V content in between. The billet was manufactured by the electric furnace process, and the 25 mm diameter rebars were reheated at 1020 °C, followed by hot-rolling at a finish rolling temperature of 950 °C. Subsequently, it was rapidly cooled by passing through a cooling zone composed of 26 coolers, and self-tempering was achieved by the latent heat of the rebar core. Detailed conditions of the Tempcore preparation process are shown in Table 2. As the V content increased, the Tempcore rebar increased in strength and decreased in elongation (Table 1).

The CCT curves for different V contents were plotted using a thermomechanical simulator (Gleeble 3500). For the Gleeble experiment, the specimens were machined into a circular shape with a diameter of 6 mm and a length of 8 mm. The specimen was heated to 980 °C at a heating rate of 10 °C/s, held for 300 s to be austenitised, and then was cooled at various rates ranging from 0.05 to 100 °C/s (0.05, 0.08, 0.1, 0.3, 0.5, 0.8, 1, 3, 5, 10, 30, 50, 80 and 100 °C/s). A holding time of 300 s was previously determined to have a grain size similar to that of prior austenite (12.4–14.5 μm) for rebars after rolling [18]. Finally, the CCT diagram was drawn using the phase transformation temperature, microstructure and hardness values.

For micro-structural observations, the cross-section was mechanically polished and etched using 3% nital. The microstructure was observed using scanning electron microscopy (SEM), and the prior austenite grain size (PAGS) was observed to confirm the grain refinement by V addition. In the case of PAGS, a special etchant was used because the microstructure is sensitive to the composition and, etching duration and temperature of the etchant. The etching method used was a mixture of 100 mL of distilled water, 4 g of picric acid + 0.2 mL of HCl, and 20 mL of surfactant at 80 °C. V precipitates were confirmed by transmission electron microscopy (TEM) images, and the specimens were prepared using the replica method. A component analysis of the precipitates was performed using energy dispersive spectroscopy (EDS). The size distribution and volume fraction of the V precipitates are quantified with approximately 150 precipitates by scanning 10 TEM micrographs for each specimen. Furthermore, the hardness of the specimens was measured using a Vickers hardness tester at 0.5 mm intervals with a 1 kgf load placed on the etched cross-section of the rebar.

### 2.2. ANN Modelling Procedure

ANN is a statistical learning algorithm that simplifies and mimics a biological nervous system such as the brain [25]. Multi-layered perceptron (MLP) is the ANN structure mostly used to determine the approximation of a function; Figure 1 shows the architecture of a typical MLP structure. An MLP consists of an input layer, a hidden layer and an output layer. The input layer receives data input from the outside and the output layer prints out the processed data (result). The hidden layer is composed of several hidden nodes, and the linear combination of values transmitted from the input layer is processed as a non-linear function and transmitted to another hidden layer or the output layer [27].

Data for the input/output layers are required to model the strength characteristics of high-strength Tempcore rebars using an ANN. In the previous studies, data were collected and adapted from the results of similar studies [19,28]. However, to increase the reliability of the results, this study used the authors’ experimental data on Tempcore rebars produced in a single plant. The input layer of the ANN model consisted of eight neurons, made up of the alloying elements (C, Mn, V, Cr, Mo, P and S) and the self-tempering temperature in the Tempcore process. The alloying elements, excluding C, Mn and V, were inevitably introduced into the scrap iron during the steelmaking process, and various ranges of V content and self-tempering temperature were used to understand the strengthening mechanism. The output layer consisted of two neurons corresponding to the yield and tensile strengths. Detailed parameters and ranges are listed in Table 3. To determine the optimal ANN model and to optimise the reproducibility, 58 datasets were divided into 43 training datasets and 15 testing datasets.

For an ANN, the learning adjusts the weights associated with each connection between neurons until the output value calculated for each input data mimics the actual output value as closely as possible. The weighted sums of the input components (
$net_{j}$
) are calculated using the following equation:$$\left(net_{j}\right)=\overset{n}{\underset{i=1}{\sum }}w_{\mathrm{ij}}x_{i}$$
where (
$net_{j}$
) is the weighted sum of the *j^th^* neuron for the input received from the preceding layer with *n* neurons,
$\;w_{\mathrm{ij}}$
is the weight of the *j^th^* neuron in the previous layer,
$x_{i}$
is the output of the *i^th^* neuron in the previous layer.

In this study, the ANN model was trained using the backpropagation algorithm, and a sigmoid function was used as the activation function. The sigmoid function is an activation function commonly used in multiple layers and is expressed as follows:$$Out_{i}=f\left(net_{j}\right)=\frac{1}{1+e^{-k\left(net_{j}\right)}}$$
where
k
is a constant used to control the slope of the semi-linear region. The sigmoid nonlinearity is activated in every layer, except in the input layer [29].

Structures with one and two hidden layers were compared to optimise the appropriate number of neurons and hidden layers for the ANN model. The average training error decreases as the total number of neurons in the hidden layer increases. For more than 18 neurons, the performance of the model saturated with no further improvement. However, the average training error in the two hidden layer structures decreased compared with that in the single hidden layer. Therefore, an ANN modelling with 18 neurons in two hidden-layer structures was used in this study.

## 3. Results and Discussion

### 3.1. CCT Diagram and Microstructure of Tempcore Rebars of various V Contents

Figure 2 shows the CCT diagrams of the rebars with various V contents. The diagram shows the cooling curve, cooling rate, transformation range, transformation microstructure and Vickers hardness. In specimen RB V2 with 0.140 wt.% V, the Ar3 temperature (781 °C) increased by 19 °C relative to specimen RB V0 (762 °C) with no V (Figure 2a,c). Therefore, the Ar3 temperature increases with V content. This is because austenite is transformed into ferrite at a relatively high temperature owing to V, a ferrite-stabilizing element [30]. The microstructures formed in the temperature range of the transformation curve are defined as ferrite, pearlite, bainite and martensite. The specimen RB V0 underwent ferrite and pearlite transformation at cooling rates in the range of 0.05–3 °C/s, and bainite began to transform when the cooling rate was higher than 1 °C/s. However, pure bainite could not be obtained under all cooling conditions. Martensite transformation appeared at a cooling rate of 5 °C/s or higher, and transformation into full martensite appeared at 50 °C/s or higher.

Figure 2b,c show CCT diagrams of specimens RB V1 and RB V2, respectively. The shapes of the diagrams are similar to that of specimen RB V0. However, in specimens RB V1 and RB V2, the cooling rate range in which ferrite and pearlite were present decreased, and bainite transformation was observed even at a lower cooling rate than that of specimen RB V0 (0.8 °C/s for RB V1 and 0.5 °C/s for RB V2). The tendency to form bainite, even at a low cooling rate, was more evident in RB V2 (with a high V content). This is because V segregates toward the austenite grain boundary, which increases the surface energy and inhibits the formation of grain boundary ferrite, leading to bainite nucleation from the austenite grain boundary [31,32]. Therefore, it is confirmed that the addition of V contributed to the formation of the low temperature microstructure of the rebars.

Figure 3 shows the microstructure of the core and surface of the rebar with various V contents. In the core of RB V0 (Figure 3a) and RB V1 (Figure 3b), ferrite (F), degenerated pearlite (DP) and bainite (B) were observed. DP is known to be formed at the boundary between the end temperature of pearlite formation and the start temperature of bainite formation, where the carbon diffusion time required for continuous layered pearlite formation is insufficient [33,34]. This is why RB V0 and RB V1 consisted of DP and B, respectively, owing to the high cooling rate of the Tempcore process. In the core of RB V2, which has the largest V content, only ferrite and bainite were observed, without DP, and the bainite fraction was higher, compared to RB V0 and RB V1. The appearance of B is consistent with the variation in the CCT diagram as the V content increases (Figure 2). Generally, the surface of a Tempcore rebar is transformed into martensite by quenching and then self-tempering by the latent heat of the core to form tempered martensite (TM) [17,18,35]. Tempered martensite, in which carbides were formed along the laths and packet boundaries, was observed on the surface of all the rebar specimens (Figure 3d–f).

The mechanical properties of tempered martensite are affected by the tempering temperature and prior austenite existing at high temperatures before its transformation to martensite. Since the tempering temperatures of the specimens used in this study were mostly ~540 °C (Table 2), the effect of V addition on the prior austenite grain size (PAGS) was considered in this study (Figure 4d). The average PAGS of RB V0, RB V1 and RB V2 were 55.9 ± 4.9, 44.6 ± 1.1 and 40.1 ± 1.5 μm, respectively, and the PAGS decreased as the V content increased. V is a strong carbide and nitride former, which exists in a solid solution of austenite at high temperatures, and precipitates as the cooling proceeds below the solid solution temperature of carbides and nitrides [36,37,38]. Therefore, the PAGS decreases with increasing V content because of the difference in the solubility of V precipitates, and the following equation gives the solubility of V precipitates as a function of the equilibrium temperature [39,40].
$$\log\left[V\right]\left[C\right]=\frac{-9500}{T}+6.72$$
$$\log\left[V\right]\left[N\right]=\frac{-8700}{T}+3.63$$
where
$\left[V\right],\;\left[C\right],\left[N\right]$
are the concentrations (wt.%) of each element and T is the absolute temperature. In the equilibrium state calculated using this formula, the complete dissolution temperatures of VC were 854 and 896 °C for RB V1 and RB V2, respectively, and the complete dissolution temperatures of VN were 988 and 1043 °C for RB V1 and RB V2, respectively. The dissolution temperature of V(C, N) has not been clearly reported, but it is known to have intermediate melting temperatures of VC and VN [40]. Therefore, in specimen RB V2, VN may not be completely dissolved at the reheating temperature (~1020 °C), so it is considered that the austenite growth is inhibited by the pinning effect of some precipitates that remain undissolved at high V contents. However, in specimen RB V1, the dissolution temperature of the precipitates was lower than the reheating temperature. Therefore, the pinning effect by the precipitates disappeared, and the austenite grains were controlled owing to the solute drag effect of the dissolved V atoms.

### 3.2. Precipitates in Tempcore Rebars with Various V Contents

Figure 5 shows an analysis of the precipitates formed due to V addition. Specimen RB V0 had a large precipitate of Fe_3_C, with no formation of V(C, N) (Figure 5a,b). In RB V1 (Figure 5d,e), elliptical V(C, N) precipitates were observed in the matrix, and some precipitates appeared along the grain boundary. The V(C, N) in specimens RB V1 and RB V2 was identified by EDS analysis (Figure 5f,i). Precipitates in specimen RB V2 with a high V content were observed in the matrix to have an elliptical shape and were finer and more numerous than those of specimen RB V1 (Figure 5d,e). V(C, N) is known to be advantageous in promoting fine and intragranular precipitation because a large amount of V is dissolved at high temperatures [41,42]; the results of the present study also showed the same precipitation tendency.

Figure 6 shows the size distribution and average size of the precipitates. In specimen RB V0, Fe_3_C precipitates with a relatively coarse distribution of 170–190 nm were observed. RB V1, with a V content of 0.072 wt.% exhibited precipitates with a size of 20–60 nm predominantly. RB V2 accounted for most precipitates with a size of 10–30 nm. With the addition of V, the number of precipitates smaller than 20 nm increases significantly. Furthermore, the average precipitate sizes of RB V0, RB V1 and RB V2 were 186.0 nm, 45.2 nm and 30.5 nm, respectively, which is consistent with the size distribution of the precipitates. These results indicate that the addition of V increased the stability of the precipitate. Previous studies reported that an increase in V content is associated with the increase in the nucleation rate of the precipitate, and the interface mismatch between nano-sized V precipitates and ferrite can be reduced to obtain the high thermal stability of the precipitate [43]. Therefore, specimen RB V0 had Fe_3_C with an average size of 186.0 nm, which was significantly larger than that of V(C, N) in RB V1 and RB V2.

### 3.3. Mechanical Properties as a Function of the Precipitates

The V precipitation behaviour shown in Figure 6 was quantitatively analysed to determine its contribution to the strengthening mechanism. Precipitation strengthening mechanism depends on the size of the precipitates, and the bypass mechanism dominates for large precipitates (diameter of ≥10 nm). Orowan proposed a formula for precipitation strengthening in which the bypass mechanism is dominant, and the Ashby–Orowan formula was improved considering the spacing and distribution between precipitates as follows [44]:$$\sigma_{p}\left(MPa\right)=\frac{5.9\sqrt{f}}{d}\cdot \ln(\frac{d}{2.5\times 10^{-4}})$$
where
f is the volume fraction of the precipitate, and d
is the average diameter in microns. The volume fraction of the precipitate was obtained from the mass fraction of precipitates ($W_{V\left(C,N\right)}$
), the density of the matrix (
$\rho_{Fe}$
), and the density of precipitates ($\rho_{V\left(C,N\right)})$.
$$f=W_{V\left(C,N\right)}\times \frac{\rho_{V\left(C,N\right)}}{\rho_{Fe}}$$

The density of the Fe matrix is 7.875 g/cm^3^ and the density of V(C, N) was calculated using the density of VN (6.30 g/cm^3^) since the C content in the V(C, N) was negligible [45]. Figure 7 shows the contribution of precipitation strengthening, calculated using the Ashby–Orowan equation. Specimen RB V2, which has a large fraction of 10–30 nm in size, has a larger contribution to precipitation strengthening than specimens RB V1 and RB V0, which are dominated by precipitates of sizes 20–60 nm and 170–190 nm, respectively. The average strength increase by precipitation strengthening was 6, 18 and 35 MPa for RB V0, RB V1 and RB V2, respectively, and the strength increase due to V precipitates in the Tempcore rebar specimens was associated with the V content (Table 1).

Vanadium has also been reported to be effective in resisting martensite softening during tempering [46]. Figure 8 shows the Vickers hardness of each specimen to confirm the softening resistance of the tempered martensite formed in the Tempcore rebar. The hardness distribution showed U-shape regardless of V content, which was related to the microstructure formation due to the continuous cooling rate varying from the surface to the core. The surface hardness of all specimens was higher than 315 HV, indicating the hardness of the tempered martensite formed at a high cooling rate (Figure 3d–f). As the cooling rate decreases toward the core, a relatively low hardness value was achieved due to the formation of F, DP or B (Figure 3a–c).

In sample RB V2, which had the highest V content, the maximum hardness value was 381 HV. The maximum difference in hardness between the surfaces of specimens RB V0 and RB V2 was 67 HV. The difference in hardness of the tempered martensite is associated with the coarsening of the precipitates during self-tempering, which is dependent on carbon diffusion. V is a strong carbonitride former, and the V(C, N) produced during the self-tempering process improves the softening resistance of tempered martensite. Thus, RB V2 had the largest surface hardness, followed by RB V1 and RB V0 [46]. Furthermore, the core hardness of specimens RB V0, RB V1 and RB V2 were 217, 252 and 282 HV, respectively, and the core hardness of the rebar increased with the V content. The maximum difference in hardness at the core was 65 HV, which was explained by the strengthening mechanism of V addition (Figure 7). This study confirmed that the addition of V to the Tempcore rebar contributed to the softening resistance of the tempered martensite and to an increase in strength and hardness through low-temperature microstructure transformation, PAGS refinement and precipitation strengthening.

### 3.4. Prediction of Tensile Properties Using ANN

The prediction of mechanical properties based on material composition is a vital goal in various industries. Because the strengthening of the Tempcore rebar according to the addition of V is a result of a complex mechanism, ANN modelling was used to predict the mechanical properties. The ANN modelling results obtained using the training and testing data are shown in Figure 9. The results of the ANN model are displayed together with a linear least-squares fitted line and R-square (R^2^) and average relative error (ARE) [23]. The R^2^ values of the ANN model for the yield and tensile strengths training datasets were 95 and 96%, and ARE values for them were 2.0% and 2.1%, respectively (Figure 9a,b). In addition, the R^2^ values for the testing dataset were 94 and 93%, ARE values were 2.4 and 2.6% (Figure 9c,d), and the predicted values obtained by the ANN model successfully reproduced the experimental results. These results show that the ANN model has a high accuracy in predicting the strength of the Tempcore rebar and generalises the relationship between the input and output parameters.

Figure 10 shows the yield and tensile strengths predicted by the ANN model for the Tempcore rebar with respect to the self-tempering temperature and V content. Irrespective of the self-tempering temperature, both the yield and tensile strengths increased with the V content. Specifically, the increasing slope of the yield and tensile strengths for various V contents, rapidly increased to 0.05 wt.% V and then constantly increased above 0.05 wt.% V. The strength increment due to the V content was calculated from the difference in the average strength at the maximum (0.200 wt.%) and minimum (0.005 wt.%) V content. Therefore, the yield strength was predicted to increase by 187 MPa and the tensile strength by 174 MPa, and the addition of V had a more significant effect on the increase in the yield strength than on the tensile strength.

Figure 11 shows the strength increment with respect to the self-tempering temperature. At a relatively low self-tempering temperature, below 520 °C, the increase in yield and tensile strengths were similar. However, for the self-tempering above 530 °C, the higher the self-tempering temperature, the more pronounced the increase in the yield strength owing to the V content. Generally, a low self-tempering temperature in the Tempcore rebar plays a dominant role in improving the tensile strength through the formation of a low-temperature microstructure [47]. Thus, at a low self-tempering temperature, the increase in tensile strength by the low-temperature microstructure and the increase in yield strength by the addition of V were similar in this study. However, as the self-tempering temperature increased above 530 °C, the strengthening mechanism by the V addition dominated and significantly improved the yield strength of the Tempcore rebars. Therefore, to manufacture a Tempcore rebar with a high yield strength, which is the purpose of this study, it is recommended to make the strengthening mechanism to be dominated by V addition at a relatively high self-tempering temperature (≥530 °C). This study successfully applied ANN modelling to optimise the V content and Tempcore process conditions for the desired rebar strength.

## 4. Conclusions

In this study, the strengthening mechanism due to V addition was considered for the development of high-strength Tempcore rebars, and the strength characteristics were predicted through ANN modelling. From the results, the following conclusions can be drawn:

(1)As the V content increased from 0.005 to 0.140 wt.%, the Ar3 temperature increased and the bainite transformation curve was observed on the CCT diagram even at a low cooling rate. Therefore, the rebar core produced by the Tempcore process was observed to have a more bainitic microstructure as the V content increased.(2)The average PAGS of specimen RB V2, which had the highest V content (0.140 wt.%) was 40.1 μm, which was significantly reduced compared with specimen RB V0 (55.9 μm). This was associated with the solubility of precipitates for various V contents: grain refinement occurred in specimen RB V2 because of the pinning effect of V (C, N), which was not completely dissolved, and the solute drag effect of the dissolved V atoms during the Tempcore process.(3)V(C, N) primarily precipitated in the matrix, and the number of fine precipitates below 20 nm increased as the V content increased. The Ashby–Orowan model successfully demonstrated that the V(C, N) precipitates contributed significantly to the strengthening mechanism (specifically the yield strength) of the Tempcore rebar.(4)The ANN model successfully predicted the yield and tensile strengths of the Tempcore rebar using the main parameters such as the V content and self-tempering temperature. The data trained by the ANN model showed a high reproducibility of over 93% of R-square and the average relative error was in the range of 2.4–2.6% with the testing data.(5)The ANN prediction results show that V contents in the range of 0.01–0.20 wt.%, are more effective in increasing the yield strength at high self-tempering temperatures ≥530 °C. This result is expected to provide outstanding guidelines for optimising the V content and Tempcore process conditions for obtaining high-strength rebars in the steel industry.

## Figures and Tables

**Figure 1 materials-15-03781-f001:**
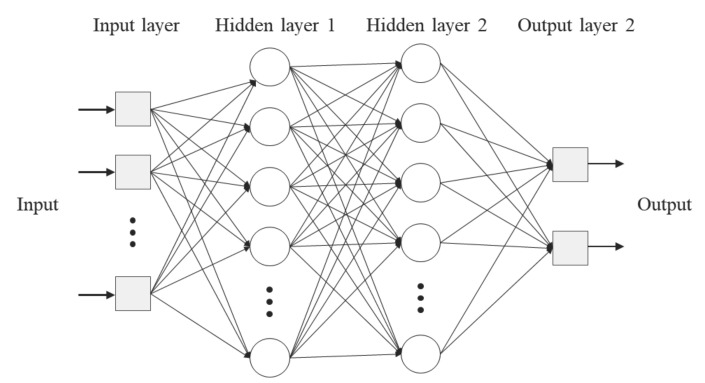
Architecture of an MLP neural network with two hidden layers.

**Figure 2 materials-15-03781-f002:**
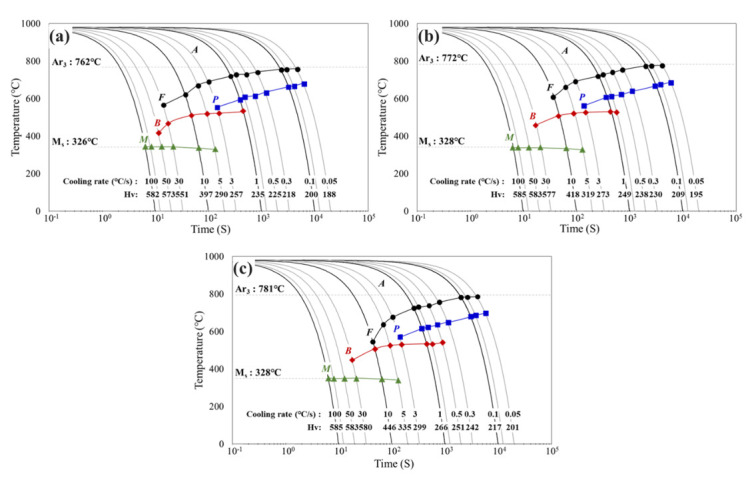
CCT diagrams drawn for various V contents: (**a**) RB V0, (**b**) RB V1, (**c**) RB V2. A−austenite, F−ferrite, P−pearlite, B−bainite and M−martensite.

**Figure 3 materials-15-03781-f003:**
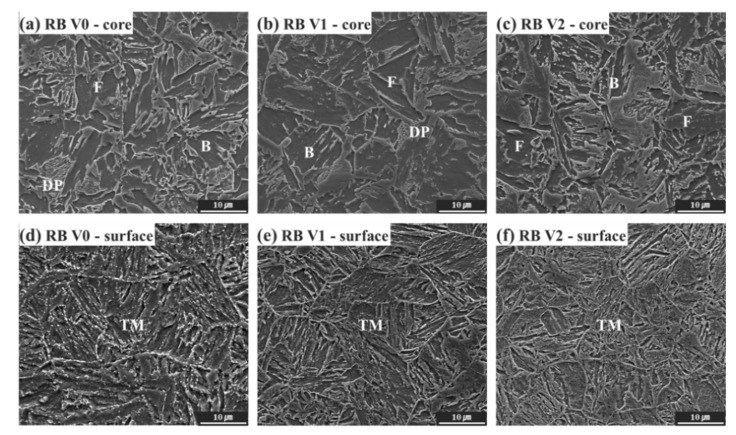
SEM micrographs of the Tempcore rebars for various locations and V contents: (**a**–**c**) in the core and (**d**–**f)** on surface; (**a**,**d**) RB Vo, (**b**,**e**) RB V1 and (**c**,**f**) RB V2; F, DP, B and TM denote ferrite, degenerated pearlite, bainite and tempered martensite, respectively.

**Figure 4 materials-15-03781-f004:**
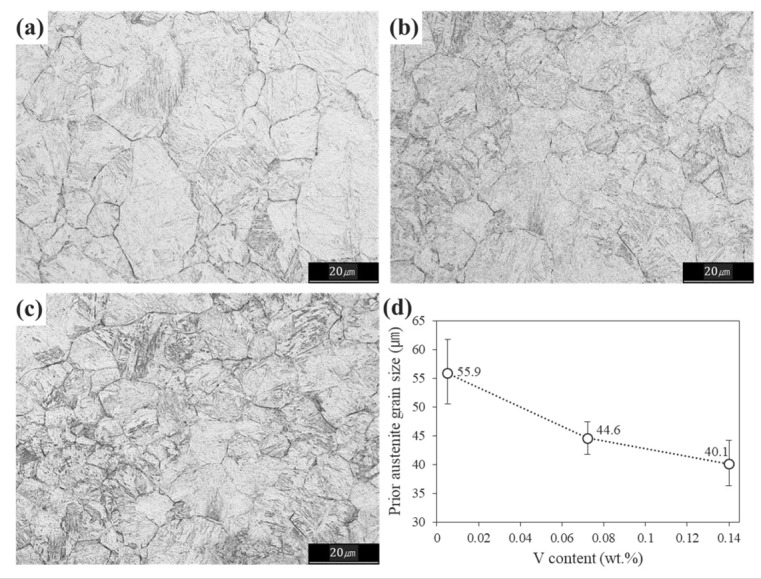
Images of the prior austenite grain and variation of PAGS with respect to V content: (**a**) RB V0, (**b**) RB V1, (**c**) RB V2, (**d**) PAGS behaviour as a function of V content.

**Figure 5 materials-15-03781-f005:**
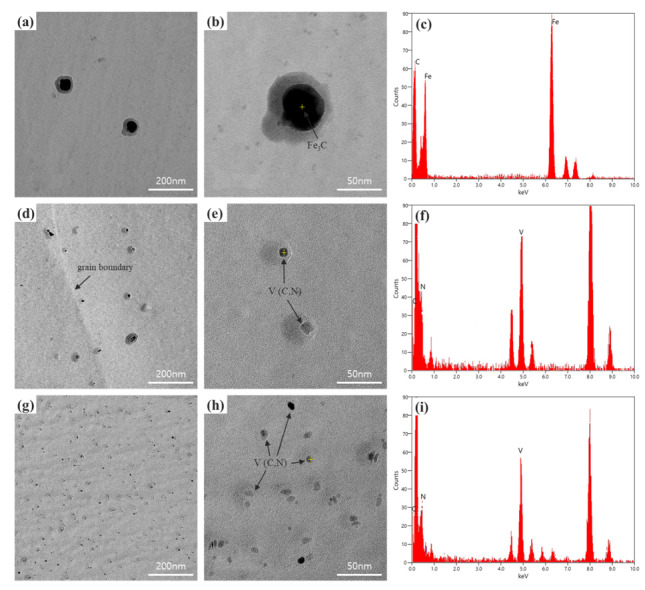
TEM images and EDS analysis of Tempcore rebar specimens showing various precipitates: (**a**,**b**,**c**) Fe_3_C for RB V0, (**d**,**e**,**f**) V(C, N) for RB V1, and (**g**,**h**,**i**) V(C, N) for RB V2.

**Figure 6 materials-15-03781-f006:**
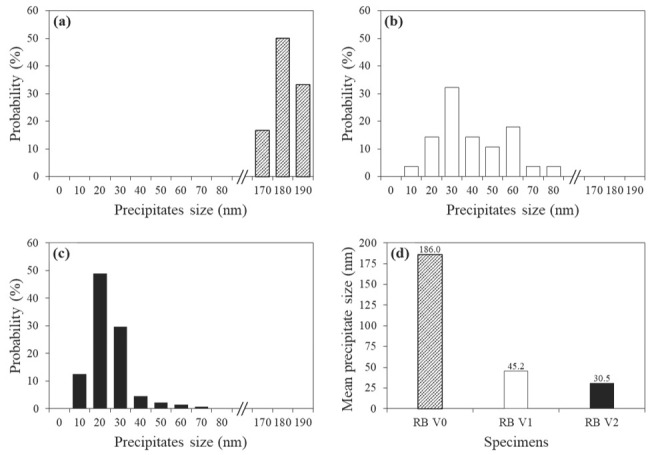
Behaviour of precipitates for various V contents; (**a**) RB V0, (**b**)RB V1, (**c**) RB V2 and (**d**) the average size of precipitates for various specimens.

**Figure 7 materials-15-03781-f007:**
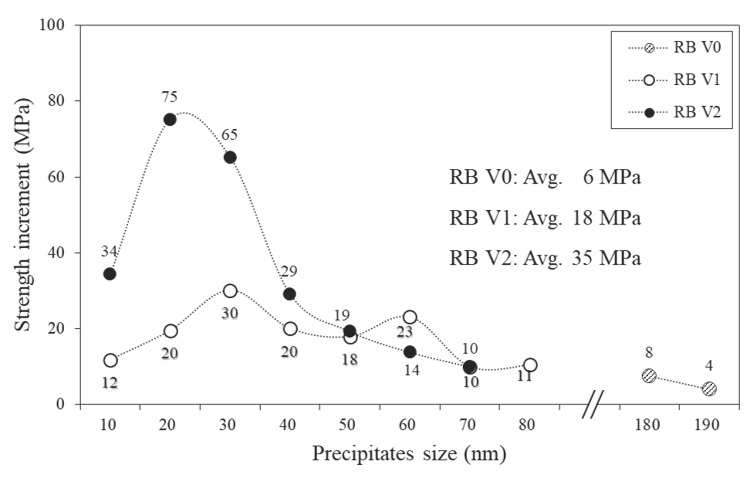
Increase in yield by precipitation hardening for various V contents.

**Figure 8 materials-15-03781-f008:**
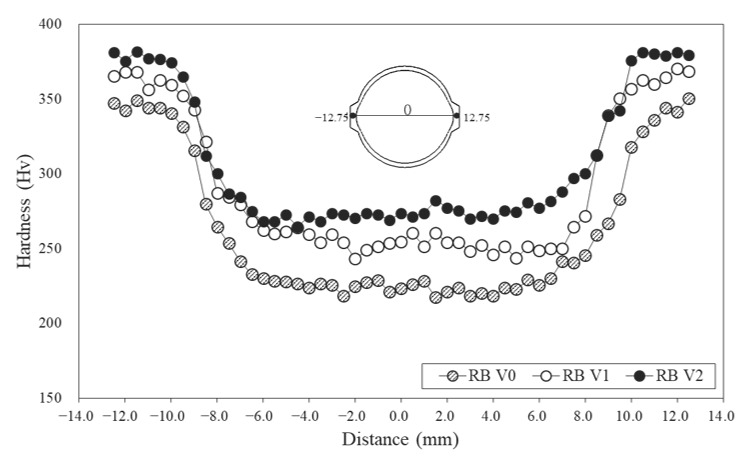
Hardness distribution of Tempcore rebars with various V contents.

**Figure 9 materials-15-03781-f009:**
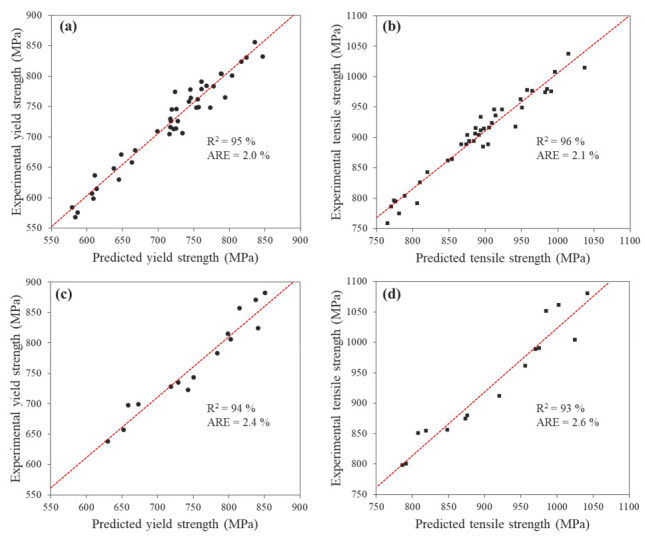
Comparison of experimental strength data with the ANN prediction lines: (**a**,**b**) yield and tensile strengths using the training set, (**c**,**d**) yield and tensile strengths using the testing set.

**Figure 10 materials-15-03781-f010:**
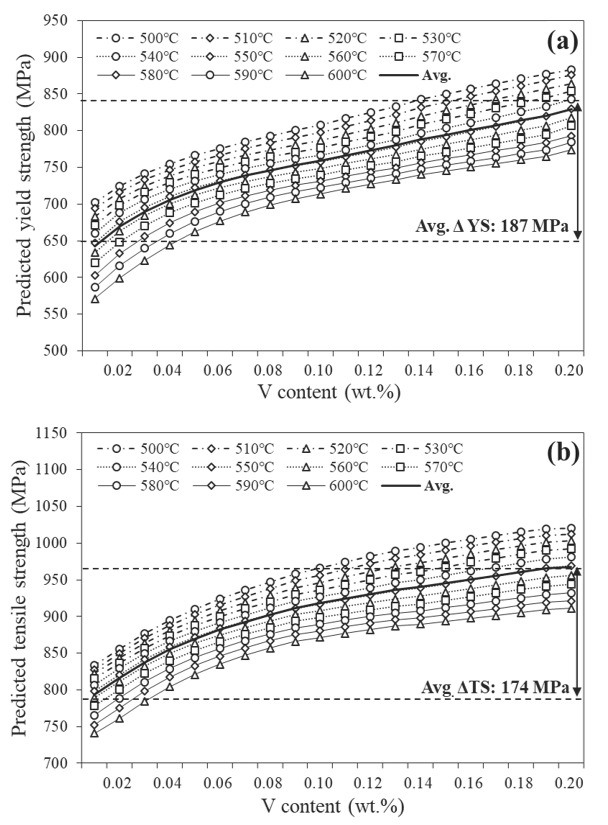
Predicted strengths against V contents for various self-tempering temperatures via ANN modelling: (**a**) yield strength and (**b**) tensile strength.

**Figure 11 materials-15-03781-f011:**
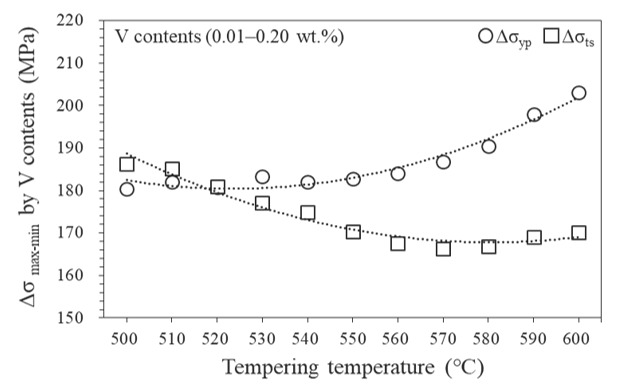
Increase in yield and tensile strengths predicted by ANN as a function of the self-tempering temperature in the regime of V contents (0.01–0.20 wt.%).

**Table 1 materials-15-03781-t001:** Chemical composition and mechanical properties of rebars.

Material	Chemical Composition (wt. %)							Mechanical Property		
	C	Mn	Si	P	S	V	Fe	YS (MPa)	TS (MPa)	El. (%)
RB V0	0.28	1.39	0.20	0.013	0.009	0.005	Bal.	652	791	13.3
RB V1	0.27	1.42	0.21	0.016	0.008	0.072	Bal.	750	904	11.6
RB V2	0.28	1.40	0.19	0.015	0.008	0.140	Bal.	796	948	9.1

**Table 2 materials-15-03781-t002:** Tempcore process conditions of rebars used in this study.

Rebar Diameter (mm)	Reheating Temp. (°C)	Finishing Roll Temp. (°C)	Quenching Time (s)	Number of the Cooler (ea.)	Self-Tempering Temp. (°C)
25	1020	980	4.3–4.8	26	540

**Table 3 materials-15-03781-t003:** Parameters and their ranges used in the ANN.

	Range	Mean	Standard Deviation
**Inputs**			
Chemical composition (wt. %)			
C	0.26–0.31	0.28	0.014
Mn	1.30–1.44	1.39	0.030
V	0.005–0.200	0.084	0.058
Cr	0.09–0.199	0.142	0.021
Mo	0.011–0.027	0.019	0.004
P	0.017–0.026	0.020	0.002
S	0.011–0.020	0.016	0.003
Tempcore process parameters (°C)			
Self–tempering temperature	501–600	1.70	30.89
**Outputs**			
Mechanical properties (MPa)			
Yield strength	579–847	720	72
Tensile strength	727–1037	880	80

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
