# Peer review of "Artificial Neural Network Modelling of the Effect of Vanadium Addition on the Tensile Properties and Microstructure of High-Strength Tempcore Rebars"

_materials, 2022, doi:10.3390/ma15113781_

Round 1

Reviewer 1 Report

  • The abstract is cursory, and a huge amount of information for this section is inappropriate. This section should explain the qualitative and quantitative findings of the investigation.
  • In section 2.1 “To confirm 78 the effect of V addition, rebars with various V contents (0.005–0.14 wt.%) were used,” regarding your work there is only three vanadium weight percentages were used so it is recommended to express the three ratios as ( 0.005, 0.072 and  14 wt.%).
  • Table 2 , “ Number of the cooler (ea.)” what mean by (ea) ????
  • Figure 2 can be represented better than this one in the manuscript, and you can use color lines
  • Figure 3 uses the definitions table inside the figure to define the letters F, B, DP, TM; hence, these are defined only in figure 2 .
  • The hardness profile in Fig.8 explains why the hardness distribution and distance take U-shape and then rechaed to a steady-like welding zone. please give an explanation for this behavior
  • Figures 9 and 10 need to be improved and the figure should be enlarged enough to be readable.
  • Figure 10 , can be separated into A and B subfigures.

Author Response

Dear reviewer,

Thank you very much for taking the time to review our paper. We also thank you for the valuable comments on this manuscript. Our point-by-point responses to the comments are detailed in the attached file.

Reviewer 2 Report

In the paper "Artificial Neural Network Modelling of the Effect of Vanadium Addition on the Tensile Properties and Microstructure of High-strength Tempcore Rebars", the authors construct the ANN-based model for the prediction of the strength of the V-contained steel. The constructed model shows high accuracy (the value of the Pearson’s coefficient is in the range of 0.93-0.96) and may be useful for the determination of the influence of the V content on the microstructure and tensile properties of the steel. The presented results seem to be interesting. The paper is well written. However, some parts of the manuscript are needed to be modified accordingly following comments:

  1. Some of the references about ANN-based models in the Introduction part are too old. It is recommended to analyze more new papers about modeling of the steels’ properties using the ANN approach (e.g., 10.3390/met12030447, 10.3390/met11050724, etc).
  2. How was chosen the number of neurons in the network? In my opinion, the number of neurons (18) is too high for such a small dataset (43 records). The authors should substantiate why a such number of neurons was used. Does it agree with a formula proposed by G. Lachtermacher and G. Fuller (Lachtermacher, G.; Fuller, J.D. Back Propagation in Time series Forecasting. J. Forecast. 1995, 14, 381–393, doi:10.1002/for.3980140405.)
  3. In the Abstract and conclusion, the authors give only the value of Pearson’s coefficient. It is better to add the absolute average relative error (please, see the formula in the uploaded file).
  4. The authors have constructed the combined machine learning model for predicting the steel properties. But it is not clear how other readers will be able to use this model? It is recommended to provide the regression equation and coefficients as Supplementary files to the manuscript.
  5. It is unclear how was measured the volume fraction of the precipitates for Ashby–Orowan formula. It is hard to measure volume fraction using TEM images due to unknown foil thickness.
  6. Minor correction:
    • error in the "Self-tempering temp" line in Table 3.

Author Response

(The authors gave the same response as above.)

Reviewer 3 Report

This work could be interesting to a specialized audience working in the same field. Nevertheless, this paper contains 11 figures, but with a little discussion. I cannot see any significant discussion of the results presented. There is no citation of the literature while authors have discussed their own results. I do not think that this paper should be published in its current form. I have a few questions and suggestions that authors of this ms should fully address.

  1. How was the hardness quantified, as discussed?
  2. Physical significance of the plot in Fig. 8 is not discussed. It is an important figure of the paper, so needs transparent discussion.
  3. In Fig. 9 itself, authors should clarify the data emerging from test and training sets. What is shown is very confusing.
  4. What causes the cross-over around 505 C in Fig. 11?
  5. The overall flow of writing must be improved. 

Author Response

(The authors gave the same response as above.)

Round 2

Reviewer 2 Report

The authors have answered on the most of previous comments and improved the manuscript. However, some replies are still questionable:

  1. The authors reply that “the optimal parameters for the network were determined based on the average training error in the output parameter prediction”. However, the number of the coefficients in the model with two hidden layers and 18 neurons is more than one hundred. This value is too large for the small dataset (43 records). The model is extremely overfitted.
  2. How was measured/calculated the mass fraction of the precipitates for calculation of the volume fraction?

Author Response

(The authors gave the same response as above.)

Reviewer 3 Report

The authors of these work have revised their paper based on my comments  and I can see there is an improvement compared to previous version Hence I recommend publication of the work. 

Author Response

Thank you for accepting this paper.